biophysics/light microscopy/image processing

algae, phase-microscopy, three-dimensional printing, open access

**Author for correspondence:**
Stephen D. Grant
e-mail: stephen.d.grant@strath.ac.uk

# Low-cost, open-access quantitative phase imaging of algal cells using the transport of intensity equation

Stephen D. Grant[1], Kyle Richford[1], Heidi L. Burdett[2,3], David McKee[1] and Brian R. Patton[1]

[1]Department of Physics and SUPA, University of Strathclyde, Glasgow G4 0NG, UK
[2]Lyell Centre for Earth and Marine Science and Technology, Edinburgh EH14 4AS, UK
[3]School of Energy, Geoscience, Infrastructure and Society, Heriot-Watt University, Edinburgh EH14 4AP, UK

SDG, 0000-0001-9186-1727; HLB, 0000-0002-3909-2235;
BRP, 0000-0001-8222-4419

Phase microscopy allows stain-free imaging of transparent biological samples. One technique, using the transport of intensity equation (TIE), can be performed without dedicated hardware by simply processing pairs of images taken at known spacings within the sample. The resulting TIE images are quantitative phase maps of unstained biological samples. Therefore, spatially resolved optical path length (OPL) information can also be determined. Using low-cost, open-source hardware, we applied the TIE to living algal cells to measure their effect on OPL. We obtained OPL values that were repeatable within species and differed by distinct amounts depending on the species being measured. We suggest TIE imaging as a method of discrimination between different algal species and, potentially, non-biological materials, based on refractive index/OPL. Potential applications in biogeochemical modelling and climate sciences are suggested.

## 1. Introduction

Phytoplankton—free-living unicellular algae found throughout the world's marine and freshwater bodies—play a pivotal role in aquatic ecosystem structure and biogeochemistry. Despite representing just 1% of the Earth's photosynthetic biomass, phytoplankton contribute an estimated 45% of global primary production (the production of organic compounds from carbon

dioxide) [1]. On evolutionary timescales, this primary production led to an increase in atmospheric oxygen concentrations, resulting in the oxygen-rich, carbon dioxide-poor atmospheric composition. This was facilitated by their role in the global carbon cycle via biomass production [2], which has also led to a proportional balance of nutrients (including nitrogen, phosphorus and trace elements) between seawater and biomass—the so-called Redfield ratio [3]. Calcifying phytoplankton, such as the coccolithophores, play additional roles in the marine carbonate cycle via the production of 'coccoliths'—calcium carbonate platelets which surround each cell [4]. Under bloom conditions, coccolithophores may contribute up to 40% of oceanic primary production and phytoplankton biomass [5]. Sinking of coccoliths that have been shed or when an algal cell dies results in a 'rain' of calcium carbonate particles through the water column, creating a transport pathway for organic and inorganic carbon from the sea surface to the seafloor [6]. The accumulation of biologically produced calcium carbonate on the seafloor (from coccolithophores as well as other calcifying marine organisms) represents the world's largest geological sink of carbon [4]. Despite these wide-ranging roles in aquatic primary production and biogeochemistry, site-specific community composition and species-specific physiological variability means that precise quantification of these roles remains poorly constrained in both marine and freshwater environments. Recent advances in measurement devices (e.g. flow cytometers) and autonomous sampling platforms (e.g. ocean gliders) have somewhat enabled this to be overcome, but remain hindered by high sensor costs, preventing replicated or networked datasets.

A key aim for monitoring marine biogeochemistry on global scales is to be able to relate ocean colour remote sensing signals to algal taxonomic and physiological properties [7]. While a significant volume of work exists attempting to derive algal functional type from remote sensing data, this effort is impeded by difficulties in the forward modelling of bulk inherent optical properties (IOPs, e.g. absorption and scattering) for algal cells. There is currently significant debate about the ability of simple Mie theory modelling, which assumes uniform spherical particles, to adequately capture the optical characteristics of complex naturally occurring particles such as phytoplankton and mineral particles. Recent work [8–10] has shown how flow cytometry can be used to establish size and bulk refractive index distributions for natural particle populations, and that these can be used to predict bulk optical and biogeochemical properties. This study used simple Mie theory, but required extrapolation of data beyond the range of observations in order to achieve closure with bulk IOPs. It has also been suggested [11] that internal structures within these particle types may be a significant contributor to the bulk optical properties. This study achieved similar levels of closure with bulk IOPs without extrapolation partical size distributions, but instead used a layered sphere model for optical scattering and made assumptions about layer thickness and refractive index distributions within the particles. Both approaches are reliant upon availability of relatively expensive equipment and are limited by relatively poor access to refractive index data for the particle population. There is a significant need to develop new tools to measure both the physical dimensions and bulk refractive index of these particle types, with mapping refractive index distributions across particles a highly desirable end-goal.

We propose a low-cost, open-access, easy to use device using microscopy techniques which would allow for morphological distinction of algal species in an adaptable, expandable platform designed to allow multiple measurement techniques to be added in a modular fashion. This small device could be incorporated into flow cytometry systems or deployed in addition to existing tools on autonomous underwater vehicles, drones, or as part of a water treatment facility monitoring system among other uses. This paper details one such proposed technique: the implementation of quantitative phase microscopy which enables refractive index extraction.

Phase microscopy allows stain-free imaging of transparent biological samples and reveals additional information and detail regarding fine structure when compared to bright field microscopy. Typical phase microscopy methods require the use of expensive optical elements or complex structured light schemes [12–18] which can be prohibitively expensive for many in educational settings or developing nations. Low-cost methods of implementing phase microscopy have been proposed using devices such as smartphones [19], however as the Raspberry Pi is an open platform this affords more robust open and low-cost access than smartphone-based platforms. This is aided by the long-term stability of the physical hardware layout and the available input–output and hardware control interfaces of the Raspberry Pi. This stability is not a consideration for smartphone manufacturers and so can lead to specific smartphone imaging methods or hardware being obsolete within very short (less than 24 months) timescales.

The transport of intensity equation (TIE), first described by Teague in 1983 [20], allows for phase microscopy to be carried out using a much simpler, and more affordable, instrumentation as it is a wholly computational, post-processing technique. The aim of this work was to assess the viability of

using the TIE to determine quantitative phase information of different strains of algae to allow for discrimination between them.

# 2. Transport of intensity equation

The TIE takes advantage of the relationship between the intensity evolution along the optical axis of a propagating wave and the wave's phase in a volume confined within discrete longitudinal steps. When an optical inhomogeneity (such as an algal cell), which affects the phase of a propagating wave, is introduced, the TIE can be used to determine the effect the resulting phase disturbance would have on the intensity of the wave after propagating through the disturbance.

Conversely, if the intensity is known along the propagation direction of the wave, then any phase objects which modulate the wave's intensity can be identified using the TIE and the phase information at the desired focal plane can be reconstructed. For sufficiently transparent samples only two images along the propagation direction of the wave are needed to obtain the intensity gradient and reconstruct the phase in the focal plane as noted in [21]. However, in this case, in line with [22] and for completeness, three images are used; one depicting the object at the focal plane of interest and two defocused images at symmetrical distances from the focal plane along the propagation path.

The calculation is carried out using the three aforementioned images, $I_1$, $I_2$ and $I_3$, where $I_2$ is the image in the focal plane and $I_1$ and $I_3$ are taken a distance $\Delta z$ away. The TIE equation is as follows:

$$-k_0 \frac{\partial I}{\partial z} = \nabla \cdot (I_0 \nabla \phi_0), \tag{2.1}$$

where $I_0$ is the intensity at the focal plane, $\phi$ is the phase, and $k_0 = (2\pi/\lambda)$ is the wavenumber. Note that $k_0$ in this instance is a general expression with the refractive index of the medium being introduced in equation (3.1) as a scaling factor. The intensity differential, $\partial I/\partial z$ is measured directly and calculated empirically from the recorded images. The introduction of an auxiliary function $\psi$ allows for the conversion of equation (2.1) into Poisson's equation. The auxiliary function is defined as

$$\nabla(\psi) = I_0 \nabla(\phi_0). \tag{2.2}$$

Simplifying equation (2.1), and substituting in equation (2.2), gives

$$-k_0 \frac{(I_3 - I_1)}{\Delta z} = \nabla^2 \psi. \tag{2.3}$$

$\Delta z$ is the experimental distance between the two out of focus images and the focal plane. $(I_3 - I_1)$ is the differential between the two out of focus intensity images.

As noted in [23], in the case of uniform illumination across the sample, the phase map can be obtained by taking the Fourier transform of equation (2.3), re-weighting its Fourier coefficients in the frequency domain, and finally performing an inverse Fourier transform to obtain the phase map in the spatial domain.

Practically, this means once the Fourier transform is calculated based on the empirically measured left-hand side of equation (2.3), the Fourier coefficients are re-weighted in accordance with equation (3) in [22], which in turn comes from equation (11) in [23]. The inverse Fourier transform is then carried out returning the now processed auxiliary function, $\psi$, which can be integrated to give the phase, $\phi$. The limiting factor on the accuracy of the TIE, and subsequent, measurements was the accuracy of the spatial offset, $\Delta z$. Complete calculation steps, along with logic to mathematically justify them, are available in the annotated python code available in the associated project repository at the DOI given in [24].

# 3. Refractive index extraction

As noted by Jesacher *et al.* [21], the 'natural' quantity returned by the TIE is optical path length (OPL). This can be obtained from the $\phi$ value obtained above by using equation (3.1). This relationship is useful in that it provides a method of determining the OPL from the phase map obtained from the TIE which will subsequently allow us to make quantitative phase measurements given certain assumptions about the sample which we will discuss further below:

$$\text{OPL} = \phi(x, y)\left(\frac{\lambda}{2\pi n_m}\right), \tag{3.1}$$

with the refractive index of the background media being, $n_m$. However, we know experimentally that the physical path length being imaged is defined by the spacing of the two out of focus images, namely $(2\Delta z)$. Therefore, values which deviate from this boundary condition correspond to changes in OPL owing to changes in refractive index rather than changes in physical path length. The OPL derived from equation (3.1), using the code we provide in the electronic supplementary material, deviates from the true OPL owing to both the Hanning Bump used to filter the frequency domain signal and the difference in refractive index between the air-based calibration of our Z stage and the actual sample refractive index. However, the fact that we need a relatively sparse sample to ensure no-cross talk between different phase objects (see further discussion in §5) means that we can recover the true OPL: regions around the object of interest provide a quantitative scaling of the OPL (sOPL) relative to the step size dz:

$$sOPL = \frac{OPL}{dz}. \tag{3.2}$$

We can calculate the excess path length (EPL) introduced by an object by subtracting the value in a region containing only the background media (sOPL) from the value in a region containing an object. This EPL is subsequently scaled by the background value. From this scaled excess path length (sEPL), the refractive index and object thickness can be determined from known values using the following expressions:

$$d = \left(\frac{n_o}{sEPL}\right) \tag{3.3}$$

and

$$n = (d_o * sEPL), \tag{3.4}$$

where sEPL is the scaled excess path length, $d_o$ is the known thickness of the object, $n_o$ is the known object refractive index relative to the background medium, and $d$ and $n$ are the calculated thickness and refractive index of the object, respectively. Therefore, if either of these values (thickness or object refractive index) are known, we can unambiguously determine the complementary value. (A note on the workflow is available in the repository [24].) This is not the case for some of the algal results shown below, but is useful in sizing particles of known refractive index (such as homogeneous sediment of known origin) or identifying the refractive index of objects of well-defined size. We present examples of both cases below to demonstrate the accuracy of this approach.

For the polymer spheres, both the diameter and the refractive index are well characterized and so both can be determined using this method. We will show with these samples that both refractive index and size is correctly returned by our methodology.

For the diamonds, the refractive index is accurately known but, while the size of individual nanodiamonds is known, they have a tendency to amalgamate together leading to clusters of diamond as shown in figure 3. Such clusters can be many times larger than individual diamonds. Using the TIE and recovering the sEPL, we can determine the size of the diamond clusters.

The algae represent a completely unknown sample as we do not know the size or refractive index accurately. In such cases, and where we want to do comparative measurements, it is important to ensure that different species are measured such that

— all samples use the same $dz$ step size; and
— the magnitude of $dz$ is chosen to be larger than all algae of interest (so that they are completely encompassed by a TIE measurement).

We emphasize that the sOPL that we show in e.g. figure 2c, is therefore potentially identical to a thicker organism with a lower refractive index. The values presented for algal cells in figure 4 are a relative optical path length (rOPL) comparing the sOPL of the cell to the sOPL of the background media. When comparing the different species in this paper, we are making an assumption of rotational symmetry (i.e. the lateral dimensions seen in the image correspond to the thickness of the organism) and, therefore, we would be able to correlate identical rOPL between species with their estimated physical dimensions. As we repeatedly emphasize, for algal identification, this approach cannot be used as a standalone technique, but is an easy to implement discrimination approach to be used in a unified workflow (i.e. to provide one yes/no answer in a chain of such questions that can lead to unambiguous identification).

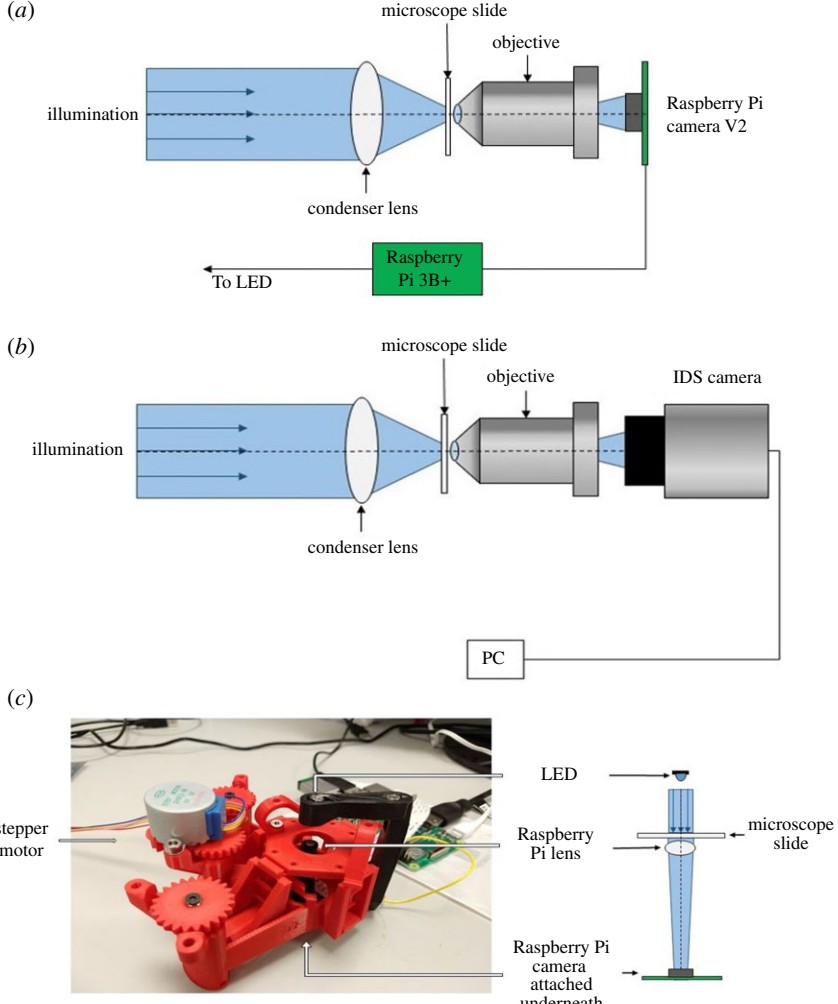

**Figure 1.** Experimental set-up showing (*a*) Raspberry Pi, and (*b*) IDS configurations laboratory-prototypes. (*c*) The openflexure scope with attached stepper motor. An LED is housed in the illumination arm (black piece), the Raspberry Pi Camera lens can be seen just below the centre of the stage, and the camera itself is mounted on the base of the unit. Ray diagrams in all schematics are illustrative only. Note: illumination for (*a*) and (*b*) was from a blue 460 nm LED, unless otherwise stated. Illumination for (*c*) was from a green 535 nm LED.

## 4. Data acquisition

We used two classes of system for the experiments performed in this paper, as shown in figure 1. To test the reliability and demonstrate the effectiveness of the TIE method, the first class of instrument consisted of low-cost components mounted on an optical table. We used the microscope in this configuration to perform a thorough set of measurements that are described fully in the results section. We refer to this system as the laboratory-prototype owing to the requirement for mounting on the optical table which provides mechanical stability and minimizes vibrations which may have adverse effects on optical alignment and imaging.

In order to show how the components used in the laboratory-prototype could be used in a device more suitable for field deployment and that has even lower infrastructure requirements, we also performed a subset of the measurements using an open source three-dimensional (3D) printable microscope designed by The OpenFlexure Project [26]. This microscope uses a Raspberry Pi camera V2 with the camera's lens removed, reversed and mounted at a greater distance from the sensor than in conventional use. The result is a medium numerical aperture system (≈0.25 calculated from the reported f# of the Raspberry Pi Camera) that can be used in place of the camera and microscope objective systems used in the laboratory-prototype. We refer to this instrument as the 'openflexure scope' in this paper. The openflexure scope used in this work ((*c*) in figure 1) was printed using an Ultimaker S5 printer and polylactic acid material. A small sample of measurements on the same algal

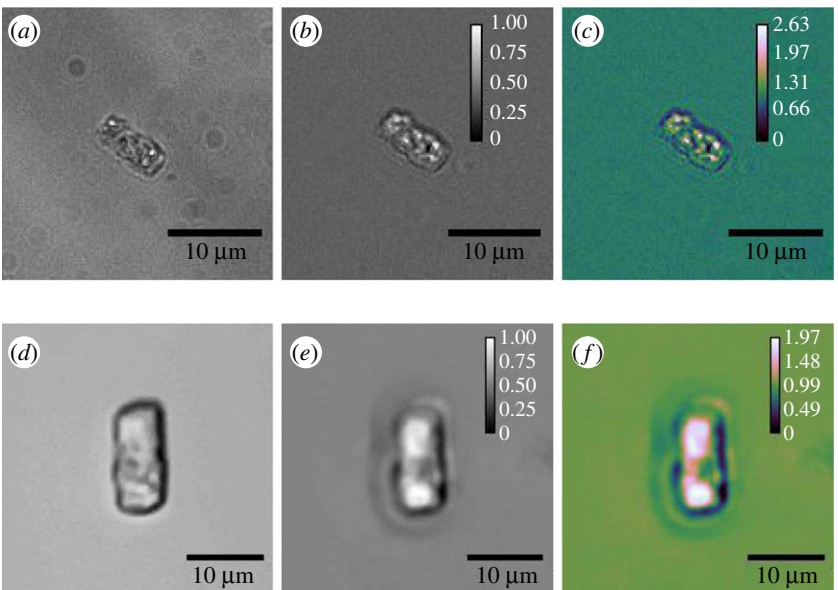

**Figure 2.** (*a*) Bright-field, focal plane image, (*b*) TIE image with normalized pixel values and (*c*) sOPL map for *Thalassiosira pseudonana* captured with the laboratory-prototype. (*d–f*) Equivalent images captured with the Raspberry Pi prototype. The sOPL map is coloured using the Cube Helix Look Up Table [25]. This colourmap is designed to better illustrate increases in intensity by taking into account perceived brightness and is useful in illustrating the relative change in sOPL from the background water value.

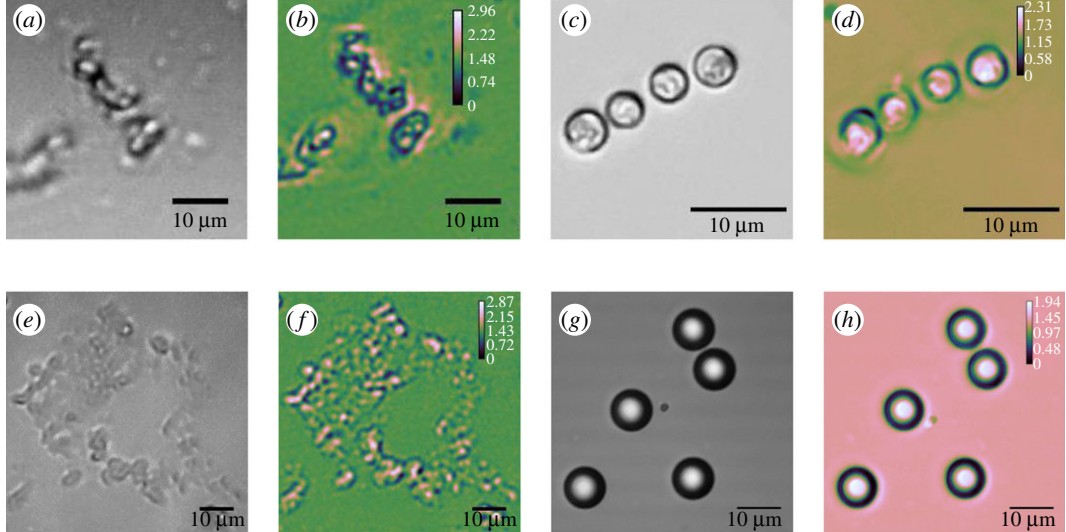

**Figure 3.** Bright-field, focal plane image, and sOPL map for (*a,b*) *T. pseudonana* captured with the openflexure scope, (*c,d*) *Emiliania huxleyi* images captured with the laboratory-prototype, (*e,f*) micro diamond samples and (*g,h*) polymer micro-spheres imaged using the Raspberry Pi prototype.

strains were captured using a green light-emitting diode (LED) for illumination. Rather than repeating all the measurements performed with the laboratory-prototype, we demonstrate that the openflexure system is equally effective once some initial characterization is performed. The results section for the openflexure scope discusses the calibration and characterization in more detail.

Figure 1*a,b* shows the laboratory-prototype apparatus. A single high brightness blue LED was used as a light source when working with these versions of the device. A low-cost Thorlabs condenser (ACL25416U: $f = 16$ mm, NA $= 0.79$) was used as the microscope condenser lens. The aqueous samples were mounted on microscope slides with an $\approx 250$ µm thick piece of electrical tape acting as a gasket to hold the liquid in place. The sample was then held vertically in a 3D printed microscope slide

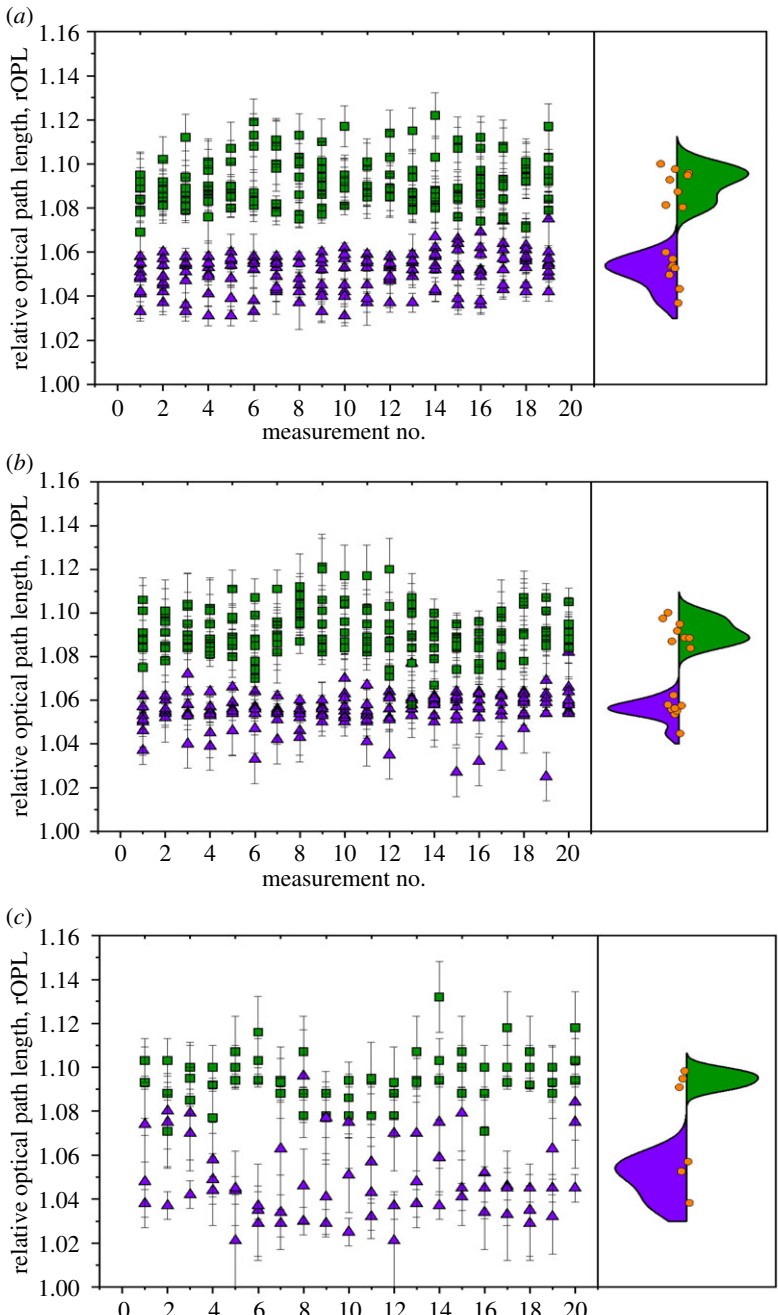

**Figure 4.** Comparison of measured samples taken with (*a*) IDS laboratory-prototype, (*b*) Raspberry Pi laboratory-prototype and (*c*) openflexure scope showing differences in measured rOPL. *Thalassiosira pseudonana* shown in purple triangles, *E. huxleyi* shown in green squares. The orange circles in the right-hand panel show the cell rOPL value for eight *T. pseudonana* cells and eight *E. huxleyi* cells (three each for the openflexure scope).

holder (designed by cfmccormick and licensed under CC BY-SA 3.0). This holder was mounted on a manufacturer calibrated Thorlabs translation stage to allow for control of the sample position relative to the objective focal plane. A Newport objective (M60-X: $m = 60x$, NA = 0.85) was used to image the sample directly. The numerical aperture mismatch between the condenser and objective led to a reduction in possible best resolution of the objective lens in the system, i.e. effectively reducing the numerical aperture of the objective to 0.79 by having less available light from the condenser.

The numerical aperture of the illumination system was estimated at 0.61 based on the condenser lens and LED used. The phase in the focal plane of the system, $\phi_0$, where we define the focal plane as being within the Rayleigh range of the excitation beam, should be consistent. The Rayleigh range was found to

be ≈0.6 mm owing to the large incident beam spot size, much larger than the step sizes of ≈10 µm used when selecting planes for the TIE calculation. The system comprising the condenser lens and objective resulted in a resolution of ≈300 nm, and a depth of field of ≈1.3 µm.

Two versions of the laboratory-prototype were tested, the sole difference between them being the imaging devices used: a UI-3060CP-M-GL camera from IDS connected to a laptop computer, and a Raspberry Pi Camera V2 connected to a Raspberry Pi 3B+. We refer to the version with the IDS camera as the 'IDS laboratory-prototype' and the Raspberry Pi version as the 'Raspberry Pi laboratory-prototype'.

The IDS camera has a sensor size of 11.345 mm × 7.126 mm with pixel size 5.86 × 5.86 µm. The Raspberry Pi camera has a sensor of dimensions 3.68 × 2.76 mm with pixel size 1.12 × 1.12 µm. The larger pixel size of the IDS camera provides a better signal to noise ratio and sensitivity. Owing to sensor dimensions, the IDS camera system had a rectangular field of view corresponding to a 200 × 100 µm region within the sample, with the Raspberry Pi camera imaging a 130 × 95 µm region in the sample.

The IDS camera was placed at the tube length of ≈160 mm from the focal plane producing a sub-Nyquist sampling rate (Nyquist sampling defined here as having pixel size of half the dimensions of the smallest resolvable features). Owing to the small sensor size on the Raspberry Pi camera, it was also operating sub-Nyquist sampling as it was placed much closer to the objective in order to obtain usable, visible images.

When using the laboratory-prototypes, images were captured at each plane by manually moving the sample using the Thorlabs translation stage with graduated micrometre with resolution of 10 µm. For the openflexure scope, a single stepper motor (a MikroElektronika 28BYJ-48) was attached on the $z$ translation adjustment gear to allow accurate control of the $z$ translation movement. This was controlled using the Raspberry Pi and an Arduino with an attached Motor Shield. The $x$ and $y$ control was carried out by hand using the two additional gears.

Illumination was controlled using python code running on attached Raspberry Pi 3B+ computers for both the laboratory-prototypes and the openflexure scope. Image acquisition with the IDS camera used proprietary IDS software connected to a laptop computer running Windows 7. Image acquisition with the Raspberry Pi laboratory-prototype and the openflexure scope was carried out using python running on the Raspberry Pi 3B+. Processing of all sets of images was carried out with a python script, available in the repository [24], and IMAGEJ [27].

Data from all versions of the device were analysed in the same manner. As the distance between out of focus planes, $2\Delta z$, was greater than the diameter of the algal cells the sOPL map needs appropriate interpretation. Owing to the difference in focal-plane and optical axis resolution, the map allows us to distinguish subcellular changes of OPL in the focal plane, but the individual values arise from the excess OPL owing to the influence of the entire cell along the optical axis at that location. Once an sOPL map was obtained from the TIE code the cells were sampled manually using the select tool in IMAGEJ. Twenty measurements of the sOPL from regions throughout the entire cell were made for each cell, analysed and then averaged to obtain an estimated sOPL value for that individual cell. Sixteen cells, eight from each of two strains, were analysed on the laboratory-prototypes, while six cells, three from each strain, were analysed with the openflexure scope.

# 5. Results

The IDS laboratory-prototype images in figure 2a–c show a bright-field image, a TIE phase image and a sOPL map of a single *Thalassiosira pseudonana* cell (CCAP 1085/12). Figure 2d–f shows equivalent images for the Raspberry Pi laboratory-prototype. Images used to create the figures and table in this work, along with python code and example raw images are available in the repository [24]. The 'shadow' effect observed in both sets of images is thought to be aberration owing to slight misalignment of the incident illumination and is more pronounced when using the Raspberry Pi camera (figure 2d–f) compared to the IDS camera (figure 2a–c).

Example images of *T. pseudonana* as captured with the openflexure scope, and *Emiliania huxleyi* (CCAP 920/12) captured with the laboratory-prototype are shown in figure 3a–d.

Figure 4 shows the measured results for each of the instruments. Figure 4a shows the 20 measurements taken with the IDS prototype for both 8 × *T. pseudonana* and 8 × *E. huxleyi* (CCAP 920/12) cells. The violin plot on the right-hand side shows the distribution of the measured rOPL values for the eight cells analysed. Figure 4b,c shows equivalent data for the Raspberry Pi

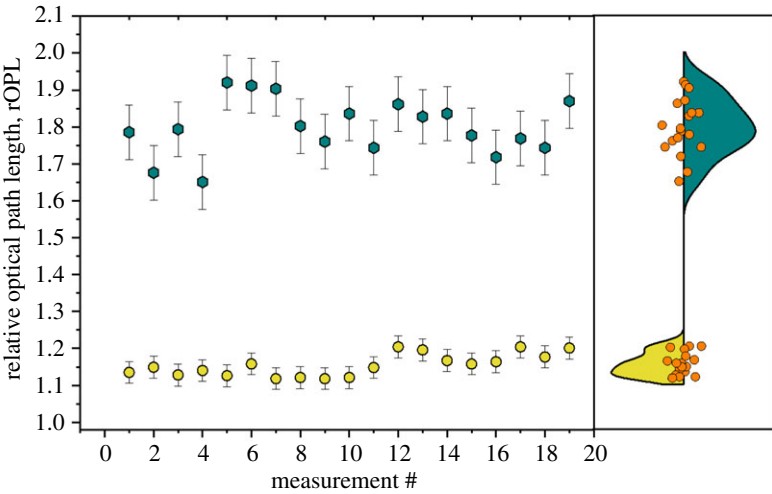

**Figure 5.** Dark cyan hexagons are micro diamond samples, while yellow circles are polystyrene micro-spheres.

**Table 1.** Mean relative optical path length for (*a*) *T. pseudonana* and (*b*) *E. huxleyi* obtained with IDS laboratory-prototype, (*c*) and (*d*) using the Raspberry Pi laboratory-prototype, and (*e*) and (*f*) using the openflexure scope.

| cell no. | (*a*) *T. pseudonana* | (*b*) *E. huxleyi* | (*c*) *T. pseudonana* | (*d*) *E. huxleyi* | (*e*) *T. pseudonana* | (*f*) *E. huxleyi* |
|---|---|---|---|---|---|---|
| 1 | 1.037 | 1.081 | 1.045 | 1.100 | 1.057 | 1.095 |
| 2 | 1.043 | 1.096 | 1.058 | 1.084 | 1.038 | 1.090 |
| 3 | 1.054 | 1.098 | 1.062 | 1.092 | 1.053 | 1.100 |
| 4 | 1.050 | 1.080 | 1.054 | 1.089 | | |
| 5 | 1.053 | 1.087 | 1.056 | 1.095 | | |
| 6 | 1.060 | 1.095 | 1.055 | 1.089 | | |
| 7 | 1.060 | 1.093 | 1.058 | 1.087 | | |
| 8 | 1.057 | 1.100 | 1.056 | 1.097 | | |
| average | 1.051 | 1.091 | 1.055 | 1.092 | 1.049 | 1.095 |
| s.d. | 0.007 | 0.006 | 0.005 | 0.005 | 0.008 | 0.003 |

laboratory-prototype and the openflexure scope, respectively. The openflexure scope data consists of six total cells as noted above.

The rOPL values found are summarized in table 1. Using the IDS laboratory-prototype, the average rOPL for *T. pseudonana* was found to be 1.051 ± 0.007, while the rOPL for *E. huxleyi* was found to be 1.091 ± 0.006. Using the Raspberry Pi laboratory-prototype, the average rOPL for *T. pseudonana* was found to be 1.055 ± 0.005, while the rOPL for *E. huxleyi* was found to be 1.092 ± 0.005; and finally, using the openflexure scope, the average rOPL for *T. pseudonana* was found to be 1.049 ± 0.008 and the rOPL for *E. huxleyi* was found to be 1.095 ± 0.003.

Verification of the method was carried out using samples of known size and refractive index. Polymer (polystyrene) micro-spheres (diameter 10 µm and refractive index 1.59) and micro diamonds (diameter 70 nm and refractive index 2.41) were imaged with the Raspberry Pi prototype. The micro-spheres were imaged at 589 nm (rather than 460 nm) using an Adafruit programmable LED in accordance with their reported refractive index of 1.59 at 589 nm. The micro diamonds were imaged using a green LED at 535 nm. Example images of both are shown in figure 3*e–h*.

As noted in §3, as the spheres are of known dimensions and refractive index both values can be returned using equations (3.3) and (3.4). Measured rOPL values for these samples are shown in figure 5. The polymer spheres had an rOPL value of 1.154 ± 0.03. The refractive index and diameter of the spheres were found to be 1.57 and 10.2 µm, respectively, compared to manufacturer provided values of 1.59 and 10 µm.

The diamond was found to have an rOPL of 1.799 ± 0.07. As the diamond only has a known refractive index, and not a known size, we can use equation (3.3) to determine the size of the clusters. Using this method, the diamond clusters were found to have a diameter of ≈1 µm.

As our samples are thin volumes on slides, they typically contain only a single-phase object in the optical path confined to a layer within the maximum and minimum range images used in the TIE. Owing to this, coupled with the fact that the measurements are relative to the surrounding medium, we are confident that within the specific parameters of this apparatus and method (numerical aperture, step magnitude, etc) these results are valid and explicit decoupling of the cell thickness is not required [19]. However, more complex or extended samples (for example a cuvette with a relatively large path length) could produce cross talk between objects outwith the plane of interest.

It is worth emphasizing that an implicit assumption in the TIE model is that the object being observed is a thin (compared to the wavelength) phase modifying object (i.e. there is low scattering or absorbtion). The test objects we use for validation nicely fit in this regime and so we see good quantitative measurement of the actual refractive index in the case of the spheres. The effectiveness of the method when characterizing particulate suspensions should, therefore, be expected to provide improved refractive index and optical characterization data to feed into remote sensing image processing models. For more complex samples, the TIE would need to be extended to properly model the propagation of a known light field through the sample. This is a significant task that would then allow accurate refractive index measurement of complex samples. We discuss below how the limitations, when properly accounted for, nevertheless allow TIE measurements to be used in quantitative discrimination tasks for algal species identification.

# 6. Discussion

The rOPL values showed a clear distinction between the two species of algae tested. The values obtained (1.051 ± 0.007 using the IDS camera and 1.055 ± 0.005 using the Raspberry Pi camera) for *T. pseudonana* and (1.091 ± 0.006 with the IDS camera and 1.092 ± 0.005 with the Raspberry Pi camera) for *E. huxleyi*, were found to show the expected distribution when compared to published refractive index values obtained from alternative methods including flow cytometry, which reports on algal values from bulk water samples falling in the range 1.05–1.15 [28]. The refractive index may also be obtained by comparing the scattering efficiency $Qb = Qc - Qa$ with theoretical scattering models. Reported values for this method at 660 nm give values of 1.035–1.063 for *T. pseudonana* and 1.093–1.095 for *E. huxleyi* [29]. As noted in [30], coccolithophores typically have higher refractive index values than those of green algae owing to their calcium carbonate shell. It is also noted that water content, metabolic condition, environmental variables, cell size, and cell age can all affect the optical properties, in particular, the refractive index, of algal cells. These factors were not closely controlled in this work but information about some of them can be gleaned from information collected using this method. Cell size can be observed directly with the microscope while environmental variables could be monitored/ controlled with the addition of temperature control hardware, illumination control etc. The addition of more robust and complete control of these factors presents an avenue for improvement and further work.

The Raspberry Pi camera, performing with pixel sizes greater than that required for Nyquist sampling, gave good agreement (within ±1%) with the more expensive and sensitive IDS equipment. Comparison of the variances in the IDS data versus the Raspberry Pi data using a two-sample F-test indicated no statistically significant difference between the methods implying comparable levels of precision. This suggests a low-cost device as proposed in the introduction could be feasible in providing useful, actionable data to allow local communities in developing nations to discriminate between types of algae, if not on a species-specific level then at a higher taxonomic level, and monitor increases in algal populations enabling precautionary action to minimize the risk of harmful algal blooms (HABs) developing, rather than reaction afterwards to mitigate the damage. The whole microscope and imaging hardware can fit in a volume of less than 1000 cm$^3$, making it portable, cheap and robust.

We have found that the TIE-based measurement approach reliably gives a narrow range of rOPL values for differing algae. However, we also suggest that this is not sufficient by itself for unambiguous determination of an algal species from rOPL data alone. The quality of imaging obtained from these compact microscopes, combined with the potential to incorporate fluorescence-based images using high-power LED's may provide a path to using multiple optical analytic techniques to narrow down potential candidates to give a warning ahead of HAB events.

The TIE allows for obtaining phase information without the need for expensive or complicated equipment. The ability to extract refractive index information provides an advantage over traditional phase microscopy methods while offering practical qualitative data from microscopy imagery. The limitations of the measurements include susceptibility to misalignment of the apparatus and assumptions regarding the background medium. These issues can potentially be overcome with the use of a 3D printed single part apparatus which fixes all optical elements in the correct position, with little to no motion allowed, reducing the potential for misalignment (while also reducing the cost of components), and the use of blank media or a medium of accurately known refractive index.

Taken as a relative measurement, this method potentially allows for discrimination of objects based on their effect on OPL against an unknown background value. Therefore, where differences in refractive index are sufficient, this low-cost, open-source, portable device may provide a useful tool, not only in relation to HABs but also in providing information regarding marine particles in general. Further development in terms of analytical capability holds the potential for enabling near real-time species identification and physiological assessment.

Data accessibility. All data related to this paper are available at the Strathclyde KnowledgeBase at the following DOI: https://doi.org/10.15129/69626e13-2fb1-4ed2-9b1b-70bfb938fe91 [24].

Authors' contributions. S.G. and K.R. carried out the experiments. H.B. provided appropriate algal samples. S.G. analysed the data. S.G. wrote the manuscript with contributions from H.B., D.M. and B.R.P. B.R.P. conceived of the work and supervised the project. All authors gave final approval for publication.

Competing interests. We declare we have no competing interests.
Funding. This work was funded under grants from the Royal Society (CHG/R1/170017 and URF/R/180017). B.R.P. holds a Royal Society University Research Fellowship.

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
