## [Reviewer comments · Royal Society Open Science]

Review History

RSOS-190921.R0 (Original submission)

Review form: Reviewer 1

Is the manuscript scientifically sound in its present form?

No

Are the interpretations and conclusions justified by the results?

No

Is the language acceptable?

Yes

Is it clear how to access all supporting data?

Yes

Do you have any ethical concerns with this paper?

No

Have you any concerns about statistical analyses in this paper?

No

Recommendation?

Major revision is needed (please make suggestions in comments)

Comments to the Author(s)

This paper demonstrates low cost ways of using phase imaging to obtain the refractive index map of algal cells. It uses the transport of intensity equation, a well known phase imaging technique that is suitable for use with standard microscopes with little modification. It provides a reasonably accurate quantitative phase map across the field of view. This phase map is then converted into a refractive index map for the algal cells. The average refractive index for different algal cells is compared in order to differentiate between them.

This paper has the potential to be useful to the community, as low cost quantitative phase microscopes are not common but have great potential for field based biological imaging. This paper's demonstration of the low-cost microscope with TIE to generate a quantitative phase image is important and could be the focus of this paper. The application in this case to obtain the refractive index of biological cells is currently flawed but has potential. A suitable comparison could be drawn with [Meng, X., Huang, H., Yan, K., Tian, X., Yu, W., Cui, H., Kong, Y., Xue, L., Liu, C. and Wang, S., 2017. Smartphone based hand-held quantitative phase microscope using the transport of intensity equation method. *Lab on a Chip*, 17(1), pp.104-109.] who develop a TIE microscope 'add-on' for a smart phone.

My major concern is how the paper attempts to decouple the refractive index from the thickness of the object using the phase image in Section 3, which is not straight forward. The main issue is in equation 3.1, which should be:

$$t(x,y) = \phi(x,y) \lambda / (2\pi \Delta n(x,y))$$

where $t(x,y)$ is the two dimensional thickness profile of an object with a constant refractive index along the z axis, and $\Delta n(x,y)$ is the difference between the refractive index of the object compared to its surrounding medium. The phase image produced by the TIE, ϕ , is not just between $-\Delta z$ and $+\Delta z$, but is the phase shift summed along the entire optical axis, from illumination to detector. The issue then arises of how much of the optical axis contains the object of unknown refractive index, $t(x,y)$. There also appears to be some confusion around the different refractive indices, with Equation 3.1 referring to nm as the sample refractive index.

I think the correct way to unlock the potential of the phase image to determine the refractive index is either to know the size of the algal cells (maybe from measurement using the in-focus image and some spherical or other approximations, or some other technique) to determine $t(x,y)$ and from that use the above equation to decouple the refractive index (using $2\Delta z$ is not correct). Alternatively, using two different wavelengths and taking two sets of measurements provides enough information to decouple. A useful review of decoupling the thickness-refractive index ambiguity is given in the introduction of [Gili Dardikman, Yoav N. Nychate, Itay Barnea, Nir A. Turko, Gyanendra Singh, Barham Javidi, and Natan T. Shaked, "Integral refractive index imaging of flowing cell nuclei using quantitative phase microscopy combined with fluorescence microscopy," *Biomed. Opt. Express* 9, 1177-1189 (2018)].

The range of values for the refractive index in Figure 3 obtained using sampling therefore doesn't take into account the thickness of the cell across its profile, as it hasn't been decoupled.

Other points:

In the abstract, phase contrast microscopy refers to a particular type of microscope. To avoid ambiguity phase imaging or phase microscopy is preferred.

The accurately measured values for Δz is not given. This is a key parameter for the setup and will be necessary for replication. In addition, it would be useful to know how the z translation of the water scope was measured, and any error in those measurements.

In Figure 2, the following improvements could be made: Crop out the unnecessary space around the cells. The phase images should have their scale normalised and displayed. It would be useful if the scale bars of the top and bottom image were the same size so that the images could be better compared with the cells the same size. The font on the scale bars should be increased. The images produced by the water scope should be included. It would also be interesting to see the images across the different algal species, to appreciate the difference between them. There appears to be some low frequency 'halo' effects in 2 (d), (e), and (f), but it is not clear whether it is because these are out of focus, or an issue with the technique.

Page 7 line 41 Δz should be $2 \Delta z$. What is the diameter of the algal cells?

The naming of the microscopes used should be consistent, the water scope is referred to as the water scope, openflexure and water microscope.

Review form: Reviewer 2 (Richard Bowman)

Is the manuscript scientifically sound in its present form?

Yes

Are the interpretations and conclusions justified by the results?

Yes

Is the language acceptable?

Yes

Is it clear how to access all supporting data?

Yes

Do you have any ethical concerns with this paper?

No

Have you any concerns about statistical analyses in this paper?

No

Recommendation?

Accept with minor revision (please list in comments)

Comments to the Author(s)

The manuscript describes measurements of the refractive index distribution of algal cells using three implementations of a simple bright-field microscope together with the Transport of Intensity equation. The authors conclude that all three of the systems under test (ranging from conventional, breadboard-style components to a much cheaper system based around a Raspberry Pi camera module and 3D printed mechanical stage) were easily able to distinguish between the two species of algae on test, and furthermore that the results produced by the different systems were not significantly different.

I've not used the TIE method for phase retrieval myself, though I have come across it in the past - I am fairly familiar with the current state of the art in low cost/open source microscopy though. I cannot, however, claim to speak with much authority on the biology in question - so I have focused on the optics/imaging/microscopy aspects of the paper.

I note that the authors have been careful to credit prior art, and are neither claiming novelty for the TIE method, or for the microscope hardware. However, they are the first team (at least to my knowledge) to combine these two techniques to get quantitative phase imaging using such inexpensive hardware. Combined with the archive of their open-source code (which I've not run, but have looked at) this makes the present work a very useful addition, as it makes maximum use of the available hardware in existing low cost microscope designs to provide an extremely useful additional imaging mode. The accompanying code is neat and well-documented and should help to make the technique attractive to other researchers.

I think the work presented is both sound and useful, and would like to see it published. I have a few technical queries which I'm sure the team can address satisfactorily. In most cases this will just represent a minor addition to the manuscript. I have deliberately written these points before accessing the material on the Strathclyde repository, as said repository is not included in the manuscript as presented for review. If said material does answer the questions, it might be nice to consider pulling in the key points to the main text if possible.

My main query regards the object's thickness. In the introduction it's stated that the thickness of the object (which is a linear scaling term in converting optical path length into refractive index) is assumed to be the distance between planes 1 and 3 (i.e. the spacing between the two images used for Transport of Intensity). However, I don't understand the basis for this assumption. Surely an object that is much thinner than this would still cause a phase shift in the transmitted light, but scaled down to reflect its shorter path length. Assuming the object extends between the planes will thus systematically underestimate its refractive index (though the recovered optical path length should be correct). The authors do point out that the "natural" quantity measured by this experiment is path length, so I am a little perplexed as to why all the plots are scaled into units of refractive index. Is it not possible that the difference in average refractive index is due to varying thickness (or possibly varying thicknesses of different materials)? It is possible I've missed something obvious here, but if so I would be delighted if the authors could correct me.

On a related note, perhaps the authors could pass comment on the effect of phase objects outside of the volume considered (i.e. above the upper plane, or below the lower plane). My guess is that such phase objects would still show up, but appear blurred. This isn't a criticism of the technique, indeed it's a feature of most if not all phase microscopy. However, it underlines the point that the assumption that the object's thickness is equal to the spacing between planes is not necessarily a good one.

There are a few more minor comments:

Page 1: "bulk IOPs without extrapolation PSDs" - I don't think PSD is defined. Particle size distribution is my guess.... This could be written more clearly I think.

Equation 2.1: I have always written vector differential equations such as this with an explicit dot or cross product between the first gradient operator and its (vector) argument. That's probably personal taste rather than convention, Teague's 1983 paper does explicitly put a dot in, while [20] does not. I'd vote for the dot, but it's a minor point.

Page 6: "the limiting factor on accuracy... was the accuracy of the spatial offset" - did the authors calibrate their translation stage and/or the Water Scope microscope to check the displacement in the focal direction was as expected, and if so, how? I think it's fine to relegate the details of the calculation to the data repository, particularly as the algorithm is already reasonably well documented in the literature, but a couple of sentences to describe how dz was calibrated (or whether it was assumed) would, I think, be warranted in the text. Particularly in the case of the micrometer stage, it would not be unreasonable to assume that the manufacturer calibrated it to some degree, but I'd like that to be explicit.

Page 6: the numerical aperture for the Raspberry Pi camera lens is given as approximately 0.25. Was this a measurement, or taken from the spec sheet? If the former, it would be useful to know as there are not many reliable measurements of this.

Page 8: the magnification of the benchtop system was set by varying the distance between the objective and the sensor. Given that it was (I assume) a finite-conjugated objective, that means that the IDS camera was operating at close to the design parameters of the lens (image formed 150mm away from the pupil) but the Raspberry Pi camera version of the system was operating quite far from those conditions. Presumably this caused some aberration in the images? A comment on that might be nice. Of course the webcam lens will also be operating away from its original design parameters, but given that it has a much lower numerical aperture anyway, I wouldn't expect this to be nearly so noticeable.

Figure 2: I assumed both sets of images were from the benchtop system using different cameras - is that the case, or are the second set of images from the system without an objective lens? I don't think the caption is explicit.

Page 12: HAB is not defined, unless I missed it. My guess was something along the lines of Harmful Algal Bloom, but this should be defined (though I don't think it affects the content of my review).

Data archive: the images taken using the Water Scope appear to be missing, but otherwise it looks good - I am impressed with the neatness and level of commenting!

General: The microscope that is referred to as Water Scope in the text looks like the OpenFlexure Microscope - there has been some confusion over the name of this project, but I believe OpenFlexure Microscope is the "official" name.

Questions aside, I enjoyed reading this and look forward to seeing it published.

Decision letter (RSOS-190921.R0)

24-Jun-2019

Dear Dr Patton:

Manuscript ID RSOS-190921 entitled "Low-cost, open-access refractive index mapping of algal cells using the transport of intensity equation" which you submitted to Royal Society Open Science, has been reviewed. The comments from reviewers are included at the bottom of this letter.

In view of the criticisms of the reviewers, the manuscript has been rejected in its current form. However, a new manuscript may be submitted which takes into consideration these comments.

Please note that resubmitting your manuscript does not guarantee eventual acceptance, and that your resubmission will be subject to peer review before a decision is made.

Your resubmitted manuscript should be submitted by 22-Dec-2019. If you are unable to submit by this date please contact the Editorial Office.

on behalf of Dr Pietro Cicuta (Subject Editor)
 openscience@royalsociety.org

Editor Comments to Author (Dr Pietro Cicuta):

Both reviewers have highlighted a possible flaw in the implementation of the TIE method, relating to the assumption that the phase path is $2\Delta z$. This needs to be considered in detail, since the bulk of the results of the paper depend on this assumption. It is possible that there is a fixed effective path difference linked to the NA of the objective? This or other assumptions need to be tested, perhaps on samples of known thickness and refractive index (polymer beads), and/or by varying the refractive index of the surrounding medium. The topic of the paper is of very valid interest, and we hope the authors can make the method work in a valid way.

Reviewers' Comments to Author:

Reviewer: 1

Comments to the Author(s)

This paper demonstrates low cost ways of using phase imaging to obtain the refractive index map of algal cells. It uses the transport of intensity equation, a well known phase imaging technique that is suitable for use with standard microscopes with little modification. It provides a reasonably accurate quantitative phase map across the field of view. This phase map is then converted into a refractive index map for the algal cells. The average refractive index for different algal cells is compared in order to differentiate between them.

This paper has the potential to be useful to the community, as low cost quantitative phase microscopes are not common but have great potential for field based biological imaging. This paper's demonstration of the low-cost microscope with TIE to generate a quantitative phase image is important and could be the focus of this paper. The application in this case to obtain the refractive index of biological cells is currently flawed but has potential. A suitable comparison could be drawn with [Meng, X., Huang, H., Yan, K., Tian, X., Yu, W., Cui, H., Kong, Y., Xue, L., Liu, C. and Wang, S., 2017. Smartphone based hand-held quantitative phase microscope using the transport of intensity equation method. *Lab on a Chip*, 17(1), pp.104-109.] who develop a TIE microscope 'add-on' for a smart phone.

My major concern is how the paper attempts to decouple the refractive index from the thickness of the object using the phase image in Section 3, which is not straight forward. The main issue is in equation 3.1, which should be:

$$t(x,y) = \phi(x,y) \lambda / (2 \pi \Delta n(x,y))$$

where $t(x,y)$ is the two dimensional thickness profile of an object with a constant refractive index along the z axis, and $\Delta n(x,y)$ is the difference between the refractive index of the object compared to its surrounding medium. The phase image produced by the TIE, ϕ , is not just between $-\Delta z$ and $+\Delta z$, but is the phase shift summed along the entire optical axis, from illumination to detector. The issue then arises of how much of the optical axis contains the object of unknown

refractive index, $t(x,y)$. There also appears to be some confusion around the different refractive indices, with Equation 3.1 referring to nm as the sample refractive index.

I think the correct way to unlock the potential of the phase image to determine the refractive index is either to know the size of the algal cells (maybe from measurement using the in-focus image and some spherical or other approximations, or some other technique) to determine $t(x,y)$ and from that use the above equation to decouple the refractive index (using $2 \Delta z$ is not correct). Alternatively, using two different wavelengths and taking two sets of measurements provides enough information to decouple. A useful review of decoupling the thickness-refractive index ambiguity is given in the introduction of [Gili Dardikman, Yoav N. Nychgate, Itay Barnea, Nir A. Turko, Gyanendra Singh, Barham Javidi, and Natan T. Shaked, "Integral refractive index imaging of flowing cell nuclei using quantitative phase microscopy combined with fluorescence microscopy," *Biomed. Opt. Express* 9, 1177-1189 (2018)].

The range of values for the refractive index in Figure 3 obtained using sampling therefore doesn't take into account the thickness of the cell across its profile, as it hasn't been decoupled.

Other points:

In the abstract, phase contrast microscopy refers to a particular type of microscope. To avoid ambiguity phase imaging or phase microscopy is preferred.

The accurately measured values for Δz is not given. This is a key parameter for the setup and will be necessary for replication. In addition, it would be useful to know how the z translation of the water scope was measured, and any error in those measurements.

In Figure 2, the following improvements could be made: Crop out the unnecessary space around the cells. The phase images should have their scale normalised and displayed. It would be useful if the scale bars of the top and bottom image were the same size so that the images could be better compared with the cells the same size. The font on the scale bars should be increased. The images produced by the water scope should be included. It would also be interesting to see the images across the different algal species, to appreciate the difference between them. There appears to be some low frequency 'halo' effects in 2 (d), (e), and (f), but it is not clear whether it is because these are out of focus, or an issue with the technique.

Page 7 line 41 Δz should be $2 \Delta z$. What is the diameter of the algal cells?

The naming of the microscopes used should be consistent, the water scope is referred to as the water scope, openflexure and water microscope.

Reviewer: 2

Comments to the Author(s)

The manuscript describes measurements of the refractive index distribution of algal cells using three implementations of a simple bright-field microscope together with the Transport of Intensity equation. The authors conclude that all three of the systems under test (ranging from conventional, breadboard-style components to a much cheaper system based around a Raspberry Pi camera module and 3D printed mechanical stage) were easily able to distinguish between the two species of algae on test, and furthermore that the results produced by the different systems were not significantly different.

I've not used the TIE method for phase retrieval myself, though I have come across it in the past - I am fairly familiar with the current state of the art in low cost/open source microscopy though. I cannot, however, claim to speak with much authority on the biology in question - so I have focused on the optics/imaging/microscopy aspects of the paper.

I note that the authors have been careful to credit prior art, and are neither claiming novelty for the TIE method, or for the microscope hardware. However, they are the first team (at least to my knowledge) to combine these two techniques to get quantitative phase imaging using such inexpensive hardware. Combined with the archive of their open-source code (which I've not run, but have looked at) this makes the present work a very useful addition, as it makes maximum use of the available hardware in existing low cost microscope designs to provide an extremely useful additional imaging mode. The accompanying code is neat and well-documented and should help to make the technique attractive to other researchers.

I think the work presented is both sound and useful, and would like to see it published. I have a few technical queries which I'm sure the team can address satisfactorily. In most cases this will just represent a minor addition to the manuscript. I have deliberately written these points before accessing the material on the Strathclyde repository, as said repository is not included in the manuscript as presented for review. If said material does answer the questions, it might be nice to consider pulling in the key points to the main text if possible.

My main query regards the object's thickness. In the introduction it's stated that the thickness of the object (which is a linear scaling term in converting optical path length into refractive index) is assumed to be the distance between planes 1 and 3 (i.e. the spacing between the two images used for Transport of Intensity). However, I don't understand the basis for this assumption. Surely an object that is much thinner than this would still cause a phase shift in the transmitted light, but scaled down to reflect its shorter path length. Assuming the object extends between the planes will thus systematically underestimate its refractive index (though the recovered optical path length should be correct). The authors do point out that the "natural" quantity measured by this experiment is path length, so I am a little perplexed as to why all the plots are scaled into units of refractive index. Is it not possible that the difference in average refractive index is due to varying thickness (or possibly varying thicknesses of different materials)? It is possible I've missed something obvious here, but if so I would be delighted if the authors could correct me.

On a related note, perhaps the authors could pass comment on the effect of phase objects outside of the volume considered (i.e. above the upper plane, or below the lower plane). My guess is that such phase objects would still show up, but appear blurred. This isn't a criticism of the technique, indeed it's a feature of most if not all phase microscopy. However, it underlines the point that the assumption that the object's thickness is equal to the spacing between planes is not necessarily a good one.

There are a few more minor comments:

Page 1: "bulk IOPs without extrapolation PSDs" - I don't think PSD is defined. Particle size distribution is my guess.... This could be written more clearly I think.

Equation 2.1: I have always written vector differential equations such as this with an explicit dot or cross product between the first gradient operator and its (vector) argument. That's probably personal taste rather than convention, Teague's 1983 paper does explicitly put a dot in, while [20] does not. I'd vote for the dot, but it's a minor point.

Page 6: "the limiting factor on accuracy... was the accuracy of the spatial offset" - did the authors calibrate their translation stage and/or the Water Scope microscope to check the displacement in the focal direction was as expected, and if so, how? I think it's fine to relegate the details of the calculation to the data repository, particularly as the algorithm is already reasonably well documented in the literature, but a couple of sentences to describe how dz was calibrated (or whether it was assumed) would, I think, be warranted in the text. Particularly in the case of the micrometer stage, it would not be unreasonable to assume that the manufacturer calibrated it to some degree, but I'd like that to be explicit.

Page 6: the numerical aperture for the Raspberry Pi camera lens is given as approximately 0.25. Was this a measurement, or taken from the spec sheet? If the former, it would be useful to know as there are not many reliable measurements of this.

Page 8: the magnification of the benchtop system was set by varying the distance between the objective and the sensor. Given that it was (I assume) a finite-conjugated objective, that means that the IDS camera was operating at close to the design parameters of the lens (image formed 150mm away from the pupil) but the Raspberry Pi camera version of the system was operating quite far from those conditions. Presumably this caused some aberration in the images? A comment on that might be nice. Of course the webcam lens will also be operating away from its original design parameters, but given that it has a much lower numerical aperture anyway, I wouldn't expect this to be nearly so noticeable.

Figure 2: I assumed both sets of images were from the benchtop system using different cameras - is that the case, or are the second set of images from the system without an objective lens? I don't think the caption is explicit.

Page 12: HAB is not defined, unless I missed it. My guess was something along the lines of Harmful Algal Bloom, but this should be defined (though I don't think it affects the content of my review).

Data archive: the images taken using the Water Scope appear to be missing, but otherwise it looks good - I am impressed with the neatness and level of commenting!

General: The microscope that is referred to as Water Scope in the text looks like the OpenFlexure Microscope - there has been some confusion over the name of this project, but I believe OpenFlexure Microscope is the "official" name.

Questions aside, I enjoyed reading this and look forward to seeing it published.

Author's Response to Decision Letter for (RSOS-190921.R0)

See Appendix A.

RSOS-191921.R0

Review form: Reviewer 1

Is the manuscript scientifically sound in its present form?

No

Are the interpretations and conclusions justified by the results?

Yes

Is the language acceptable?

Yes

Do you have any ethical concerns with this paper?

No

Have you any concerns about statistical analyses in this paper?

No

Recommendation?

Accept with minor revision (please list in comments)

Comments to the Author(s)

All of the minor points have been met completely and are appreciated. In particular the new images presented in figures 2 and 3 are greatly improved.

The major issue of decoupling the refractive index from the thickness of the object having measured the phase shift (as was noted in the previous review) has not been addressed correctly in section 3, which has not changed since the previous version.

I appreciate the work done to verify the method using known samples, but it still does not address the fundamental issue that objects with the same refractive index but different thicknesses will have different phase measurements. Either the manuscript needs to make this explicitly clear in the assumptions made, or should report the measurements as phase shifts (the natural measurement of TIE, as opposed to scaling to 'relative' refractive indices), which shouldn't change the outcome of the results, but will be a more accurate representation of the results.

Once these points have been addressed I believe this paper will be a valuable addition.

Review form: Reviewer 2 (Richard Bowman)

Is the manuscript scientifically sound in its present form?

Yes

Are the interpretations and conclusions justified by the results?

Yes

Is the language acceptable?

Yes

Do you have any ethical concerns with this paper?

No

Have you any concerns about statistical analyses in this paper?

No

Recommendation?

Accept with minor revision (please list in comments)

Comments to the Author(s)

The authors have done a great job of addressing the comments with one exception - I still struggle to understand how the sample thickness was picked. The addition of known samples (beads and micro-diamonds) is very compelling, but I couldn't determine how the object's thickness was calculated. If the objects have known thickness and refractive index, and the known thickness is used in calculating the refractive index profile, this validation experiment does not address the problem of scaling optical path length into refractive index. Instead, it simply verifies that the path length is recovered correctly. A statement of the dz values used for these experiments should be added. If, on the other hand, the known thickness was not supplied as part of the phase recovery, I would very much like to see whether the algorithm correctly determined size as well as refractive index, and to understand how this was done.

At the end of section 4, the manuscript states "the individual values arise from the mean refractive index of the entire cell along the optical axis at that location" but I think it's important

to note that the mean is taken along the whole distance from $-dz$ to $+dz$, and thus if the cell is not exactly $2dz$ in thickness, the average will include not only the cell, but quite a bit of the medium too. I don't think the authors disagree with this statement, but the manuscript definitely gives a confusing impression, by using refractive index rather than optical path length throughout.

This doesn't affect the validity of their conclusions - they still have a way of producing phase images that can distinguish between species. However, I think it's important to know whether the refractive index values have been averaged together with the refractive index of the medium or not. Note that simply specifying optical path length would get rid of this quibble completely.

The note in the data archive about Z calibration for the OpenFlexure Microscope is very helpful, and answers completely my question about how that was calibrated. To make this a useful protocol for others, it might be worth mentioning that the refractive index matters when calibrating Z: for a given translation of the sample, I would expect the focal plane to shift further, by a factor of the refractive index. That means that if the coverslips are stacked on top of each other, a different result should be obtained to the case where the coverslips are offset, creating a stepped structure where the beam does not pass through any glass. As the experiments are performed in water, I assume, I might expect a scaling factor either way.

Decision letter (RSOS-191921.R0)

29-Nov-2019

Dear Dr Grant,

On behalf of the Editor, I am pleased to inform you that your Manuscript RSOS-191921 entitled "Low-cost, open-access refractive index determination of algal cells using the transport of intensity equation" has been accepted for publication in Royal Society Open Science subject to minor revision in accordance with the referee suggestions. Please find the referees' comments at the end of this email.

The reviewers and Subject Editor have recommended publication, but also suggest some minor revisions to your manuscript. Therefore, I invite you to respond to the comments and revise your manuscript.

- Ethics statement

- Data accessibility

If you wish to submit your supporting data or code to Dryad (<http://datadryad.org/>), or modify your current submission to dryad, please use the following link:
<http://datadryad.org/submit?journalID=RSOS&manu=RSOS-191921>

- **Competing interests**

- **Authors' contributions**

- **Acknowledgements**

- **Funding statement**

Because the schedule for publication is very tight, it is a condition of publication that you submit the revised version of your manuscript before 08-Dec-2019. Please note that the revision deadline will expire at 00.00am on this date. If you do not think you will be able to meet this date please let me know immediately.

Kind regards,

on behalf of Professor Pietro Cicuta (Subject Editor)
openscience@royalsociety.org

Editor Comments to Author:

The paper has been much improved but both reviewers point out that there is still an incorrect explanation of the averaging of refractive index over the optical path. This needs to be corrected before acceptance of the manuscript.

Reviewer comments to Author:

Reviewer: 1
Comments to the Author(s)

All of the minor points have been met completely and are appreciated. In particular the new images presented in figures 2 and 3 are greatly improved.

The major issue of decoupling the refractive index from the thickness of the object having measured the phase shift (as was noted in the previous review) has not been addressed correctly in section 3, which has not changed since the previous version.

I appreciate the work done to verify the method using known samples, but it still does not address the fundamental issue that objects with the same refractive index but different thicknesses will have different phase measurements. Either the manuscript needs to make this explicitly clear in the assumptions made, or should report the measurements as phase shifts (the natural measurement of TIE, as opposed to scaling to 'relative' refractive indices), which shouldn't change the outcome of the results, but will be a more accurate representation of the results.

Once these points have been addressed I believe this paper will be a valuable addition.

Reviewer: 2

Comments to the Author(s)

The authors have done a great job of addressing the comments with one exception - I still struggle to understand how the sample thickness was picked. The addition of known samples (beads and micro-diamonds) is very compelling, but I couldn't determine how the object's thickness was calculated. If the objects have known thickness and refractive index, and the known thickness is used in calculating the refractive index profile, this validation experiment does not address the problem of scaling optical path length into refractive index. Instead, it simply verifies that the path length is recovered correctly. A statement of the dz values used for these experiments should be added. If, on the other hand, the known thickness was not supplied as part of the phase recovery, I would very much like to see whether the algorithm correctly determined size as well as refractive index, and to understand how this was done.

At the end of section 4, the manuscript states "the individual values arise from the mean refractive index of the entire cell along the optical axis at that location" but I think it's important to note that the mean is taken along the whole distance from -dz to +dz, and thus if the cell is not exactly 2dz in thickness, the average will include not only the cell, but quite a bit of the medium too. I don't think the authors disagree with this statement, but the manuscript definitely gives a confusing impression, by using refractive index rather than optical path length throughout.

This doesn't affect the validity of their conclusions - they still have a way of producing phase images that can distinguish between species. However, I think it's important to know whether the refractive index values have been averaged together with the refractive index of the medium or not. Note that simply specifying optical path length would get rid of this quibble completely.

The note in the data archive about Z calibration for the OpenFlexure Microscope is very helpful, and answers completely my question about how that was calibrated. To make this a useful protocol for others, it might be worth mentioning that the refractive index matters when calibrating Z: for a given translation of the sample, I would expect the focal plane to shift further, by a factor of the refractive index. That means that if the coverslips are stacked on top of each other, a different result should be obtained to the case where the coverslips are offset, creating a stepped structure where the beam does not pass through any glass. As the experiments are performed in water, I assume, I might expect a scaling factor their either way.

Author's Response to Decision Letter for (RSOS-191921.R0)

See Appendix B.

Decision letter (RSOS-191921.R1)

20-Dec-2019

Dear Dr Grant,

It is a pleasure to accept your manuscript entitled "Low-cost, open-access quantitative phase imaging of algal cells using the transport of intensity equation" in its current form for publication in Royal Society Open Science. The comments of the reviewer(s) who reviewed your manuscript are included at the foot of this letter.

on behalf of Dr Pietro Cicuta (Subject Editor)
openscience@royalsociety.org

Appendix A

We are happy to see the generally positive response to our paper, and would in particular like to highlight the reviewers' statements that "This paper has the potential to be useful to the community, as low cost quantitative phase microscopes are not common but have great potential for field based biological imaging." And "I think the work presented is both sound and useful, and would like to see it published."

We will now address specific points raised individually. To help, comments from the editors and reviewers are *in italic*, while our responses are in **red text**.

Editor Comments to Author (Dr Pietro Cicuta):

Both reviewers have highlighted a possible flaw in the implementation of the TIE method, relating to the assumption that the phase path is $2\Delta z$. This needs to be considered in detail, since the bulk of the results of the paper depend on this assumption. It is possible that there is a fixed effective path difference linked to the NA of the objective? This or other assumptions need to be tested, perhaps on samples of known thickness and refractive index (polymer beads), and/or by varying the refractive index of the surrounding medium.

The topic of the paper is of very valid interest, and we hope the authors can make the method work in a valid way.

We would like to thank the editor for noting the positive comments from the reviewers. We understand their concerns and feel that they are not specifically relevant for the applications we anticipated when writing the paper. However, in light of a desire to make the paper more generally applicable, we have performed the additional experiments requested, and added an additional section, beginning on page 7, in Section 5, that is as follows:

Verification of the method was carried out using samples of known size and refractive index. Polymer (polystyrene) micro-spheres (diameter 10 μm and refractive index 1.59) and micro diamonds (diameter 1 μm and refractive index 2.4) were imaged with the Raspberry Pi prototype. The micro-spheres were imaged at 589 nm (rather than 460 nm) using an Adafruit programmable LED in accordance with their reported refractive index of 1.59 @ 589 nm. The micro diamonds were imaged using a blue LED at 460 nm as all other samples on the lab and Raspberry Pi prototypes had been. Example images of both are shown in figure 3 (e - h).

Measured relative refractive index values for these samples are shown in figure 1. Diamond was measured to have a relative refractive index of 1.799 ± 0.03 giving an absolute refractive index of 2.41, while the polystyrene spheres had a relative value of 1.154 ± 0.07 , giving an absolute value of 1.545 compared to the reported value of 1.59.

As our samples are thin volumes on slides they typically contain only a single phase object in the optical path confined to a layer within the maximum and minimum range images used in the TIE. Due to this, coupled with the fact that the measurements are relative to the surrounding medium, we are confident that within the specific parameters of this apparatus and method (numerical aperture, step magnitude, etc) these results are valid. However, more complex or extended samples (for example a cuvette with a relatively large path length) could produce cross talk between objects outwith the plane of interest.

It is worth emphasising that an implicit assumption in the TIE model is that the object being observed is a thin (compared to the wavelength) phase modifying object (i.e. there is low scattering or absorption). The test objects we use for validation nicely fit in this regime and so we see good quantitative measurement of the actual refractive index. The effectiveness of the method when

characterising particulate suspensions should therefore be expected to provide improved RI and optical characterisation data to feed into remote sensing image processing models. For more complex samples, the TIE would need to be extended to properly model the propagation of a known light field through the sample. This is a significant task that would then allow accurate refractive index measurement of complex samples. We discuss below how the limitations, when properly accounted for, nevertheless allow TIE measurements to be used in quantitative discrimination tasks for algal species identification.

Reviewer: 1

Comments to the Author(s)

This paper demonstrates low cost ways of using phase imaging to obtain the refractive index map of algal cells. It uses the transport of intensity equation, a well known phase imaging technique that is suitable for use with standard microscopes with little modification. It provides a reasonably accurate quantitative phase map across the field of view. This phase map is then converted into a refractive index map for the algal cells. The average refractive index for different algal cells is compared in order to differentiate between them.

This paper has the potential to be useful to the community, as low cost quantitative phase microscopes are not common but have great potential for field based biological imaging. This paper's demonstration of the low-cost microscope with TIE to generate a quantitative phase image is important and could be the focus of this paper. The application in this case to obtain the refractive index of biological cells is currently flawed but has potential. A suitable comparison could be drawn with [Meng, X., Huang, H., Yan, K., Tian, X., Yu, W., Cui, H., Kong, Y., Xue, L., Liu, C. and Wang, S., 2017. Smartphone based hand-held quantitative phase microscope using the transport of intensity equation method. Lab on a Chip, 17(1), pp.104-109.] who develop a TIE microscope 'add-on' for a smart phone.

A citation for the above paper has been added with a note on a difference in approach due to the openness of the platforms used.

My major concern is how the paper attempts to decouple the refractive index from the thickness of the object using the phase image in Section 3, which is not straight forward. The main issue is in equation 3.1, which should be:

$$t(x,y) = \phi(x,y) \lambda / (2\pi \Delta n(x,y))$$

where $t(x,y)$ is the two dimensional thickness profile of an object with a constant refractive index along the z axis, and $\Delta n(x,y)$ is the difference between the refractive index of the object compared to its surrounding medium. The phase image produced by the TIE, ϕ , is not just between $-\Delta z$ and $+\Delta z$, but is the phase shift summed along the entire optical axis, from illumination to detector. The issue then arises of how much of the optical axis contains the object of unknown refractive index, $t(x,y)$. There also appears to be some confusion around the different refractive indices, with Equation 3.1 referring to nm as the sample refractive index.

We do not disagree with the reviewer on this point in general, however the TIE works by measuring the refractive index gradient from sets of images, and so for objects far outside the plane of interest, there will be no discernible contribution to the phase imaging, as the rate of change of the intensity across the images will be indistinguishable from any noise in the images. Therefore, there is a practical range within which we are detecting phase changes which will be influenced by the actual hardware in use in the experiment. We discuss some implications of this in the verification section.

I think the correct way to unlock the potential of the phase image to determine the refractive index is either to know the size of the algal cells (maybe from measurement using the in-focus image and some spherical or other approximations, or some other technique) to determine $t(x,y)$ and from that use the above equation to decouple the refractive index (using $2\Delta z$ is not correct). Alternatively, using two different wavelengths and taking two sets of measurements provides enough information to decouple. A useful review of decoupling the thickness-refractive index ambiguity is given in the introduction of [Gili Dardikman, Yoav N. Niygate, Itay Barnea, Nir A. Turko, Gyanendra Singh, Barham

Javidi, and Natan T. Shaked, "Integral refractive index imaging of flowing cell nuclei using quantitative phase microscopy combined with fluorescence microscopy," *Biomed. Opt. Express* 9, 1177-1189 (2018)].

A citation to this paper has been added.

The range of values for the refractive index in Figure 3 obtained using sampling therefore doesn't take into account the thickness of the cell across its profile, as it hasn't been decoupled.

Added section addressing verification using known size and RI samples which addresses the points above.

We agree that our nomenclature wasn't sufficiently precise, and therefore highlight that the effective refractive index we recover relates to the mean refractive index measured through a column of the cell along the optical propagation axis. See final paragraph of section 3 for details.

Other points:

In the abstract, phase contrast microscopy refers to a particular type of microscope. To avoid ambiguity phase imaging or phase microscopy is preferred.

Removed 'contrast' from any mentions of 'phase contrast microscopy', leaving 'phase microscopy' as a more general term.

The accurately measured values for Δz is not given. This is a key parameter for the setup and will be necessary for replication. In addition, it would be useful to know how the z translation of the water scope was measured, and any error in those measurements.

The Lab prototype and Raspberry Pi prototype used a translation stage that was manufacturer calibrated and assumed valid. The steps were taken as 10 μm according to the micrometre. Based on correct recovery of the RI of known samples as outlined in the verification section, we concluded that this assumption was indeed valid.

Notes on calibration of openflexure scope z translation are added to the data archive.

In Figure 2, the following improvements could be made: Crop out the unnecessary space around the cells. The phase images should have their scale normalised and displayed. It would be useful if the scale bars of the top and bottom image were the same size so that the images could be better compared with the cells the same size. The font on the scale bars should be increased. The images produced by the water scope should be included. It would also be interesting to see the images across the different algal species, to appreciate the difference between them. There appears to be some low frequency 'halo' effects in 2 (d), (e), and (f), but it is not clear whether it is because these are out of focus, or an issue with the technique.

Edited figure 2 to crop images, add normalised calibration bars, adjusted scale bars, and clarified caption.

Added example images, figure 3, from openflexure scope and of other species and samples.

Added note on aberration:

"The 'shadow' effect observed in both sets of images is thought to be aberration due to slight misalignment of the incident illumination and is more pronounced when using the Raspberry Pi camera (images d-f) compared to the IDS camera (images a-c)."

Page 7 line 41 Δz should be $2\Delta z$. What is the diameter of the algal cells?

Added '2' to 'dz'.

On average the diameter of the spherical *E. huxleyi* was ~4 μm , while the cylindrical *T. pseudonana* were ~10 μm in length and ~5 μm in diameter. These dimensions were roughly homogeneous within a single sample of cells.

The naming of the microscopes used should be consistent, the water scope is referred to as the water scope, openflexure and water microscope.

Changed all mentions of 'water scope' or 'flexure scope' to 'openflexure scope' for consistency

Reviewer: 2

Comments to the Author(s)

The manuscript describes measurements of the refractive index distribution of algal cells using three implementations of a simple bright-field microscope together with the Transport of Intensity equation. The authors conclude that all three of the systems under test (ranging from conventional, breadboard-style components to a much cheaper system based around a Raspberry Pi camera module and 3D printed mechanical stage) were easily able to distinguish between the two species of algae on test, and furthermore that the results produced by the different systems were not significantly different.

I've not used the TIE method for phase retrieval myself, though I have come across it in the past - I am fairly familiar with the current state of the art in low cost/open source microscopy though. I cannot, however, claim to speak with much authority on the biology in question - so I have focused on the optics/imaging/microscopy aspects of the paper.

I note that the authors have been careful to credit prior art, and are neither claiming novelty for the TIE method, or for the microscope hardware. However, they are the first team (at least to my knowledge) to combine these two techniques to get quantitative phase imaging using such inexpensive hardware. Combined with the archive of their open-source code (which I've not run, but have looked at) this makes the present work a very useful addition, as it makes maximum use of the available hardware in existing low cost microscope designs to provide an extremely useful additional imaging mode. The accompanying code is neat and well-documented and should help to make the technique attractive to other researchers.

I think the work presented is both sound and useful, and would like to see it published. I have a few technical queries which I'm sure the team can address satisfactorily. In most cases this will just represent a minor addition to the manuscript. I have deliberately written these points before accessing the material on the Strathclyde repository, as said repository is not included in the manuscript as presented for review. If said material does answer the questions, it might be nice to consider pulling in the key points to the main text if possible.

We would like to note that a freely accessible repository was included in the manuscript as submitted for review.

My main query regards the object's thickness. In the introduction it's stated that the thickness of the object (which is a linear scaling term in converting optical path length into refractive index) is assumed to be the distance between planes 1 and 3 (i.e. the spacing between the two images used for Transport of Intensity). However, I don't understand the basis for this assumption. Surely an object that is much thinner than this would still cause a phase shift in the transmitted light, but scaled down to reflect its shorter path length. Assuming the object extends between the planes will

thus systematically underestimate its refractive index (though the recovered optical path length should be correct). The authors do point out that the “natural” quantity measured by this experiment is path length, so I am a little perplexed as to why all the plots are scaled into units of refractive index. Is it not possible that the difference in average refractive index is due to varying thickness (or possibly varying thicknesses of different materials)? It is possible I’ve missed something obvious here, but if so I would be delighted if the authors could correct me.

As Reviewer 1 raised similar points, we would like to thank both reviewers for highlighting a weakness in our language. We think that the changes in response to the request for validation data along with the discussion on the meaning of the effective refractive index will clarify this point and answer the reviewer appropriately.

On a related note, perhaps the authors could pass comment on the effect of phase objects outside of the volume considered (i.e. above the upper plane, or below the lower plane). My guess is that such phase objects would still show up, but appear blurred. This isn’t a criticism of the technique, indeed it’s a feature of most if not all phase microscopy. However, it underlines the point that the assumption that the object’s thickness is equal to the spacing between planes is not necessarily a good one.

Added section addressing verification using known size and RI samples which addresses the above points. See also our response to reviewer 1’s similar point.

There are a few more minor comments:

Page 1: “bulk IOPs without extrapolation PSDs” - I don’t think PSD is defined. Particle size distribution is my guess.... This could be written more clearly I think.

Defined PSD as particle size distribution.

Equation 2.1: I have always written vector differential equations such as this with an explicit dot or cross product between the first gradient operator and its (vector) argument. That’s probably personal taste rather than convention, Teague’s 1983 paper does explicitly put a dot in, while [20] does not. I’d vote for the dot, but it’s a minor point.

Added dot.

Page 6: “the limiting factor on accuracy...was the accuracy of the spatial offset” - did the authors calibrate their translation stage and/or the Water Scope microscope to check the displacement in the focal direction was as expected, and if so, how? I think it’s fine to relegate the details of the calculation to the data repository, particularly as the algorithm is already reasonably well documented in the literature, but a couple of sentences to describe how dz was calibrated (or whether it was assumed) would, I think, be warranted in the text. Particularly in the case of the micrometer stage, it would not be unreasonable to assume that the manufacturer calibrated it to some degree, but I’d like that to be explicit.

Added note that stage was calibrated by the manufacturer and assumed valid.

Page 6: the numerical aperture for the Raspberry Pi camera lens is given as approximately 0.25. Was this a measurement, or taken from the spec sheet? If the former, it would be useful to know as there are not many reliable measurements of this.

Added that the NA of Rasp pi camera was calculated from f#.

Page 8: the magnification of the benchtop system was set by varying the distance between the objective and the sensor. Given that it was (I assume) a finite-conjugated objective, that means that the IDS camera was operating at close to the design parameters of the lens (image formed 150mm away from the pupil) but the Raspberry Pi camera version of the system was operating quite far from those conditions. Presumably this caused some aberration in the images? A comment on that might be nice. Of course the webcam lens will also be operating away from its original design parameters, but given that it has a much lower numerical aperture anyway, I wouldn't expect this to be nearly so noticeable.

Added note on aberration:

"The 'shadow' effect observed in both sets of images is thought to be aberration due to slight misalignment of the incident illumination and is more pronounced when using the Raspberry Pi camera (images d-f) compared to the IDS camera (images a-c)."

Figure 2: I assumed both sets of images were from the benchtop system using different cameras - is that the case, or are the second set of images from the system without an objective lens? I don't think the caption is explicit.

Clarified caption on figure 2 to label which images go with which system.

Page 12: HAB is not defined, unless I missed it. My guess was something along the lines of Harmful Algal Bloom, but this should be defined (though I don't think it affects the content of my review).

Defined HAB as harmful algal blooms

Data archive: the images taken using the Water Scope appear to be missing, but otherwise it looks good - I am impressed with the neatness and level of commenting!

Data archive is up to date.

General: The microscope that is referred to as Water Scope in the text looks like the OpenFlexure Microscope - there has been some confusion over the name of this project, but I believe OpenFlexure Microscope is the "official" name.

Changed all mentions of 'water scope' or 'flexure scope' to 'openflexure scope' for consistency

Appendix B

To whom it may concern,

Please find our response to editor and reviewer comments below.

Yours faithfully,

Stephen Grant, on behalf of all authors

Editor Comments to Author:

The paper has been much improved but both reviewers point out that there is still an incorrect explanation of the averaging of refractive index over the optical path. This needs to be corrected before acceptance of the manuscript.

Authors:

We note that the reviewers have again commented on our use of "relative refractive index". Our inclusion of calibration objects of known refractive index and/or size was added to demonstrate that, when used appropriately, the technique is fully quantitative and returns correct values for either the object size or refractive index, when the other quantity is known. We acknowledge that the fact that both reviewers have highlighted this means that perhaps our language is not precise enough and so in the spirit of cooperation we have amended section 3 to more fully explain both the different use scenarios for different quantities returned, and which assumptions are required when using the TIE algorithm in these cases. We have also changed to title to better reflect the reviewer's comments on optical path length (rather than refractive index) being the natural quantity returned by the TIE algorithm. In line with suggestions we have amended the paper to include use of optical path length values rather than refractive index values while expanding upon this to include further discussion of validation methods.

Reviewer comments to Author:

Reviewer: 1

Comments to the Author(s)

All of the minor points have been met completely and are appreciated. In particular the new images presented in figures 2 and 3 are greatly improved.

The major issue of decoupling the refractive index from the thickness of the object having measured the phase shift (as was noted in the previous review) has not been addressed correctly in section 3, which has not changed since the previous version.

Authors:

Section 3 has been updated to include further steps and assumptions used and to add terminology to clarify the method.

I appreciate the work done to verify the method using known samples, but it still does not address the fundamental issue that objects with the same refractive index but different thicknesses will have different phase measurements. Either the manuscript needs to make this explicitly clear in the assumptions made, or should report the measurements as phase shifts (the natural measurement of TIE, as opposed to scaling to 'relative' refractive indices), which shouldn't change the outcome of the results, but will be a more accurate representation of the results.

Once these points have been addressed I believe this paper will be a valuable addition.

Authors:

The values have been relabelled as relative optical path lengths, rather than relative refractive indices to clarify this and avoid confusion. We also highlight how the ambiguity between thickness and relative refractive index may occur and discuss how we think that this ambiguity will not arise in general use.

Reviewer: 2

Comments to the Author(s)

The authors have done a great job of addressing the comments with one exception - I still struggle to understand how the sample thickness was picked. The addition of known samples (beads and micro-diamonds) is very compelling, but I couldn't determine how the object's thickness was calculated. If the objects have known thickness and refractive index, and the known thickness is used in calculating the refractive index profile, this validation experiment does not address the problem of scaling optical path length into refractive index. Instead, it simply verifies that the path length is recovered correctly. A statement of the dz values used for these experiments should be added. If, on the other hand, the known thickness was not supplied as part of the phase recovery, I would very much like to see whether the algorithm correctly determined size as well as refractive index, and to understand how this was done.

Authors:

Section 3 and the results section have both been expanded to address these issues and terminology has been changed to help clarify the physical quantities measured. We have added a better description both of our workflow and when we feel confident in deriving refractive index or object size, and, when not, how the phase information that is returned can nevertheless lead to useful information on the sample.

At the end of section 4, the manuscript states "the individual values arise from the mean refractive index of the entire cell along the optical axis at that location" but I think it's important to note that the mean is taken along the whole distance from -dz to +dz, and thus if the cell is not exactly 2dz in thickness, the average will include not only the cell, but quite a bit of the medium too. I don't think the authors disagree with this statement, but the manuscript definitely gives a confusing impression, by using refractive index rather than optical path length throughout.

Authors:

The terminology has been changed to more accurately reflect the measured quantities with refractive index replaced with relative optical path length (rOPL). We also highlight in section three some key assumptions that must be met in order to allow reliable comparison of the phase information derived from different species.

This doesn't affect the validity of their conclusions - they still have a way of producing phase images that can distinguish between species. However, I think it's important to know whether the refractive index values have been averaged together with the refractive index of the medium or not. Note that simply specifying optical path length would get rid of this quibble completely.

The note in the data archive about Z calibration for the OpenFlexure Microscope is very helpful, and answers completely my question about how that was calibrated. To make this a useful protocol for

others, it might be worth mentioning that the refractive index matters when calibrating Z: for a given translation of the sample, I would expect the focal plane to shift further, by a factor of the refractive index. That means that if the coverslips are stacked on top of each other, a different result should be obtained to the case where the coverslips are offset, creating a stepped structure where the beam does not pass through any glass. As the experiments are performed in water, I assume, I might expect a scaling factor their either way.

Authors:

We comment on this calibration and scaling effect when discussing the derivation of the excess path length in section 3.